# Accurate Computation of Airfoil Flow Based on the Lattice Boltzmann Method

**Liangjun Wang [1], Xiaoxiao Zhang [1], Wenhao Zhu [1], Kangle Xu [2], Weiguo Wu [3], Xuesen Chu [4] and Wu Zhang [1,5,\*]**

[1] School of Computer Engineering and Science, Shanghai University, Shanghai 200444, China; shu_wlj@shu.edu.cn (L.W.); 7758721@shu.edu.cn (X.Z.); whzhu@i.shu.edu.cn (W.Z.)

[2] Shanghai Aircraft Design and Research Institute, Shanghai 201206, China; xukangle@comac.cc

[3] Department of Computer Science and Technology, Xi'an Jiaotong University, Xi'an 710049, China; wgwu@xjtu.edu.cn

[4] National Supercomputing Center in Wuxi, Wuxi 214072, China; cxs503@163.com

[5] Shanghai Institute of Applied Mathematics and Mechanics, Shanghai 200444, China

[\*] Correspondence: wzhang@shu.edu.cn; Tel.: +86-021-56331451

**Abstract:** The lattice Boltzmann method (LBM) is an important numerical algorithm for computational fluid dynamics. This study designs a two-layer parallel model for the Sunway TaihuLight supercomputer SW26010 many-core processor, which implements LBM algorithms and performs optimization. Numerical experiments with different problem sizes proved that the proposed model has better parallel performance and scalability than before. In this study, we performed numerical simulations of the flows around the two-dimensional (2D) NACA0012 airfoil, and the results of a series of flows around the different angles of attack were obtained. The results of the pressure coefficient and lift coefficient were in good agreement with those in the literature.

**Keywords:** parallel computing; Pflops supercomputer; LBM; NACA airfoil

---

## 1. Introduction

With the development of the supercomputer theory and computer hardware technology, numerical computation has become an important technical mechanism in scientific work. Computational fluid dynamics (CFD) is a significant research field. Among the many numerical methods in the CFD field, the lattice Boltzmann method (LBM) [1,2] is a numerical simulation method with great potential. It is derived from a lattice gas automaton, which is based on the microscopic thermal motion in fluid molecules. The statistical average describes mesoscopically the macroscopic motion of the fluid. The method treats fluid as a mass of small particles with only the mass and no volume. These small particles move on a regular grid, collide with surrounding particles, and transfer statistical data. The movement of a large number of particles helps in obtaining macroscopic properties of the fluid, such as density, velocity, and pressure.

The collision migration idea of LBM elementary particles, and the use of Cartesian mesh division, enable simple boundary condition processing and natural parallelism, which is suitable for large-scale numerical computation on the supercomputer [3]. Ulrich Rüde et al. presented a novel multi-physics coupled algorithm based on the LBM, which employs an Eulerian description of fluid and ions, combined with a Lagrangian representation of moving charged particles, and achieved excellent performance and scaling on up to 65,536 cores of a current supercomputer [4]. For large-scale grids, Manfred Krafczyk et al. discussed modeling and computational aspects of their approach, and presented computational results, including experimental validations [5]. G Wellein et al. introduced the

lattice Boltzmann benchmark kernels, a suite for benchmarking simple LBM kernels, which may be used for performance experiments or can act as a blueprint for an implementation [6]. Hager et al. analyzed the performance and energy to solution properties of a lattice Boltzmann flow solver on the chip and highly parallel levels for an Intel Sandy Bridge (SNB) EP (Xeon E5-2680)-based system [7]. Obrecht et al. introduced a decomposition approach for generic three-dimensional (3D) stencil problems with formulations for calculating dynamically copied position indexes, subdomain addresses, subdomain size, and halo cells, and the new pipelined 3D LBM code outperforms the original OpenCL version by 33%, by overlapping computation and communication [8]. Lintermann et al. proposed a new robust algorithm to automatically generate hierarchical Cartesian meshes on distributed multi-core HPC(High Performance Computing) systems with multiple levels of refinement [9]. Song Liu et al. proposed an effective approach to accelerate the LBM computing by fully exploiting temporal locality on a shared memory multi-core platform [10].

The supercomputer research community of China is already ranked among the top in the world. China's supercomputer has ranked first in the TOP500 ranking 10 consecutive times, from June 2013 to November 2017. In the TOP500 list in November 2018, China's Sunway TaihuLight and Tianhe-2A supercomputers ranked third and fourth, respectively [11]. Sunway TaihuLight is also the first system with a peak performance of more than 100 PFlops in the world. It uses the SW26010 many-core processor, which contains a management processing element (MPE) and 64 computing processing elements (CPEs)—a core-group (CG)—and one node has four CGs, totaling 260 processing units. The Sunway TaihuLight has a total of 40,960 SW26010 processors with superior computing power. Based on the Sunway TaihuLight, "10M-core scalable fully-implicit solver for nonhydrostatic atmospheric dynamics" and "nonlinear earthquake simulation on Sunway TaihuLight" won the highest award in the field of high-performance computing—the Gordon Bell Award [12]—in 2016 and 2017.

In terms of aeronautical-engineering computations, the wing is the main device for generating lift. Its structure is complex, and has become the focus of research [13,14]. The numerical simulation of the two-dimensional (2D) airfoil simplifies the complexity of wind tunnel experiments and obtains better experimental results. It is the primary means of designing various high-lift airfoil types. Among the various airfoils, the most representative is the National Advisory Committee for Aeronautics (NACA) airfoil series [15]. This series of airfoils has been used in the aviation field for many years and has good aerodynamics performance. In this series, NACA0012 has become the actual standard for numerical method research and verification owing to its symmetry and appropriate relative thickness.

## 2. The Lattice Boltzmann Method

The LBM can be seen as a special discrete format for solving continuous Boltzmann equations [16,17]. In 1954, Bhatnagar, Gross, and Krook proposed the famous Bhatnagar–Gross–Krook (BGK) collision model [18], which replaced the nonlinear term in the original Boltzmann equation with a simple collision operator, and it guaranteed some conditions, such as mass momentum and energy conservation. According to the BGK collision model, the lattice Boltzmann equation with the discretization of velocity and time can be obtained as the evolution equation of the particle:

$$f_\alpha(x + e_\alpha \delta_t, t + \delta_t) - f_\alpha(x,t) = -\frac{1}{\tau}\Big[f_\alpha(x,t) - f_\alpha^{eq}(x,t)\Big] + \delta_t F_\alpha(x,t), \tag{1}$$

where $\tau$ is the relaxation time and $f_\alpha^{eq}$ is the equilibrium distribution function. According to the DnQm series model proposed by Qian et al. [19] (where $n$ and $m$ are respectively represented as the spatial dimension and the discrete velocity number), the equilibrium distribution function is unified and expressed as

$$f_\alpha^{eq} = \rho \omega_\alpha [1 + \frac{e_\alpha \cdot u}{c_s^2} + \frac{(e_\alpha \cdot u)^2}{2c_s^4} - \frac{u^2}{2c_s^2}], \tag{2}$$

where $c_s = \sqrt{RT}$ is the lattice sound velocity and the value is $\frac{1}{\sqrt{3}}$, $u$ is the velocity, $\omega_\alpha$ is the weight coefficient, and $e_\alpha$ is the direction vector. In this paper, the D2Q9 model is used (Figure 1), and the parameters are selected as follows:

$$\omega_i = \begin{cases} 4/9, \ e_i^2 = 0 \\ 1/9, \ e_i^2 = c^2 \\ 1/36, \ e_i^2 = 2c^2 \end{cases}$$

$$e = c\left( \begin{array}{ccccccccc} 0 & 1 & 0 & -1 & 0 & 1 & -1 & -1 & 1 \\ 0 & 0 & 1 & 0 & -1 & 1 & 1 & -1 & -1 \end{array} \right)$$

where $c = \delta x / \delta t, \delta x, \delta t$ are the grid step size and time step, respectively. Velocity and density can be obtained from the following formula:

$$\rho = \sum_\alpha f_\alpha^{eq}, \rho u = \sum_\alpha e_\alpha f_\alpha^{eq}.$$

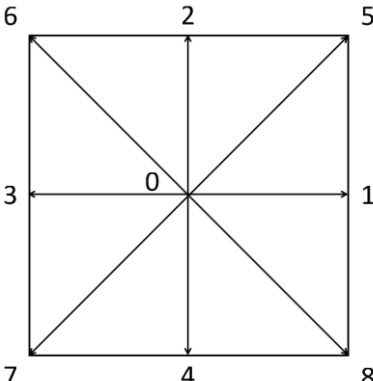

**Figure 1.** The two dimensional and nine velocity (D2Q9) model.

## 3. Computation Model

### 3.1. Generation of Airfoil, Judgment of Grid Point Type

As shown in Figure 2, we select the NACA0012 airfoil as the computation object. The airfoil consists of a series of line segments connected by scatter points. After partitioning the Cartesian grid, a known point A is determined inside the airfoil, and, in turn for each grid point, a type judgment is made. For the grid point B to be judged, it is only necessary to calculate whether the line segment AB intersects the line segment constituting the airfoil. If it does intersect, point B is then considered an external point, i.e., it is a fluid point. If it does not intersect, then point B is an internal point, i.e., a solid point. After finding all solid points, a circle of grid points around the solid point is considered the boundary point.

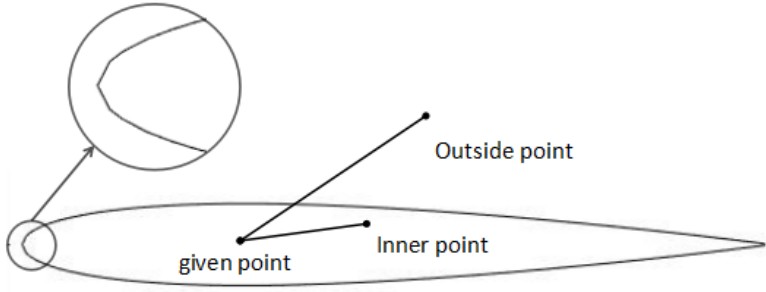

**Figure 2.** NACA0012 airfoil grid point type judgment diagram.

### 3.2. Data Partition and Communication

When using MPI (Message Passing Interface) to accelerate the program in parallel, there are two ways to partition the mesh, namely, one-dimensional (1D) partitioning and 2D partitioning, as shown in Figure 3a,b. According to the literature [20], 2D partitioning is more advantageous than 1D partitioning. Therefore, this paper adopts the 2D partitioning method; at each peripheral boundary of each MPI process, a layer of data buffer is set to store the boundary data transmitted by the adjacent process. The computation area of the original size is $NX \times NY$, and the size with the buffer is changed to $(NX + 2) \times (NY + 2)$ ($NX$ and $NY$ are the number of grids in the horizontal and vertical directions, respectively).

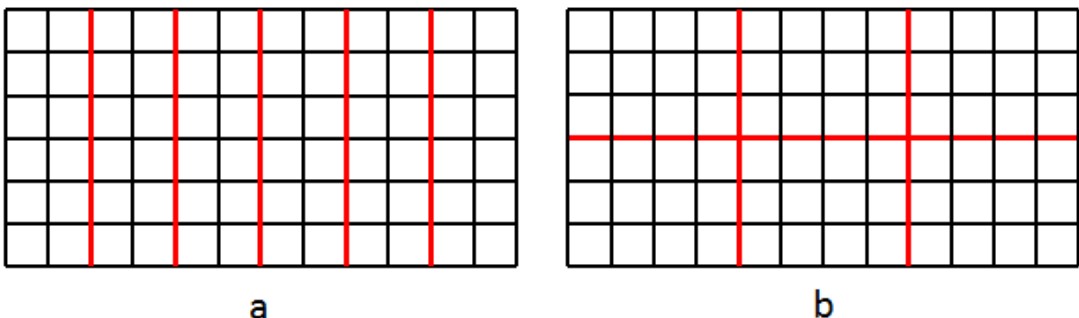

**Figure 3.** Data partitioning methods: (**a**) one-dimensional (1D) partitioning; (**b**) two-dimensional (2D) partitioning.

### 3.3. Many-Core Structure and Communication of the Sunway TaihuLight

In the SW26010 heterogeneous many-core processor, each CG contains one MPE and one operational core array, which consists of 64 CPEs, an array controller, and a secondary instruction cache. Therefore, the parallelism of MPI exists only between the MPEs, and the computational tasks on each MPE need to be further divided into CPEs.

Since each CPE has only 64 K of memory, the MPE cannot transfer the calculated data to the CPEs at one time. Instead, the MPE transfers part of the calculated data to the CPEs after the CPEs computation is completed, and the data is written into the MPE's memory. Then, we proceed to the processing of the next part of the computation data (Figure 4a). However, this will result in frequent data communication between the MPE and CPEs, which increases the time cost. In order to solve this problem, this paper adopts a double-buffer mechanism (Figure 4b, Table 1). Each time the computation data is transmitted, about half of the memory is reserved from the core. The next data are transmitted while the data are being calculated on the CPEs. After the computation is complete, we check if the data transmission is completed; if it is completed, the next computation is performed. The computation of one data part at the same time starts the transmission of the next data part at the same time, and, if it is not completed, the CPEs wait. The double-buffer model can effectively reduce most of the communication time to improve the computation efficiency.

For the computation of the grid point migration part, the information of the eight neighboring grid points around the current grid point is required. However, the neighboring grid point is discretely stored in the MPEs memory and cannot be transferred to the CPEs memory by direct memory access (DMA) at one time; therefore, it leads to multiple accesses to the MPEs memory and a higher time cost. To solve this problem, MPEs adjust the grid point data of the current grid point's neighbor from discrete storage to continuous storage before the data are transmitted between the MPEs and CPEs. The MPEs are responsible for adjusting the storage each time when the data are being computed by the CPEs, and, therefore, no additional time is required (as shown in Figure 5).

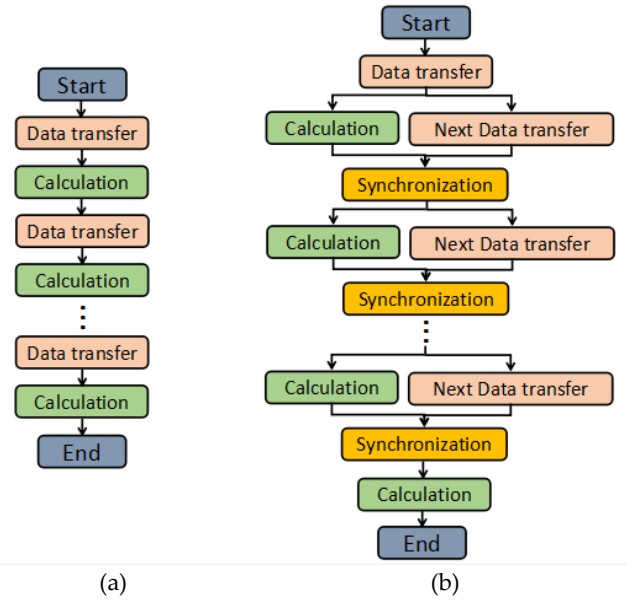

**Figure 4.** (**a**) General data transfer mode; (**b**) Double cache data transfer mode.

**Table 1.** Pseudo-code of double cache data transfer mode.

```
… …
do k = kmin, kmax
if (mod((k-kmin),corenum)+1.eq.slavecore_id) then
j = jmin
index = mod((j-jmin),2) + 1
put_reply(mod((j-jmin)+1,2)+1)=1
! Read in the first batch of data
get_reply(index)=0
call athread_get(0,a(imin,j,k), a_slave(imin,index), (imax-imin+1)*4, get_reply(index), 0, 0, 0)
call athread_get(0,b(imin,j,k), b_slave(imin,index), (imax-imin+1)*4, get_reply(index), 0, 0, 0)

do j = jmin,jmax
index = mod((j-jmin),2)+1
next = mod((j-jmin)+1,2)+1
last = next
! Read in the data needed for the next round of calculation
if (j.lt.jmax) then
get_reply(next)=0
call athread_get(0,a(imin,j+1,k),a_slave(imin,next), (imax-imin+1)*4,get_reply(next),0,0,0)
call athread_get(0,b(imin,j+1,k),b_slave(imin,next), (imax-imin+1)*4,get_reply(next),0,0,0)
endif

do while (get_reply(index).ne.2)
enddo ! Wait for the data required for this round of calculation to be read in
do i = imin, imax
c_slave(i,index)=a_slave(i,index)*a_slave(i,index)+b_slave(I,index)*b_slave(i,index)
enddo

put_reply(index)=0
call athread_put(0,c_slave(imin,index),c(imin,j,k),(imax-imin+1)*4,put_reply(index),0,0)
do while (ptu_reply(last).ne.1)
Enddo
! waiting for the last round of data to be written back
! the first round does not have to wait for the direct pass.
enddo
do while (ptu_reply(index).ne.1)
enddo ! waiting for the last batch of data to be written back
endif
enddo
… …
```

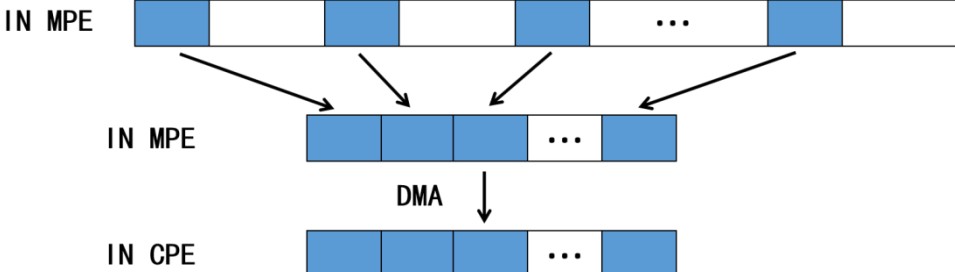

**Figure 5.** Data storage order adjustment.

In order to further reduce the communication time, we divide the data in each MPE into the form shown in Figure 6, which are internal data and boundary data, respectively, where boundary data is the part that needs to communicate with other processes. When CPEs conduct internal data computation, the MPE is responsible for communicating the boundary data with related processes.

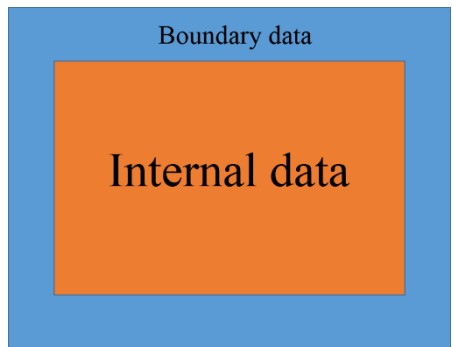

**Figure 6.** Boundary communication hiding.

In addition to the double-buffering mode, we can further reduce the communication time by using register communication between CPEs, as shown in Figure 7 (only the first line CPE is displayed, lines 2 to 8 are similar).

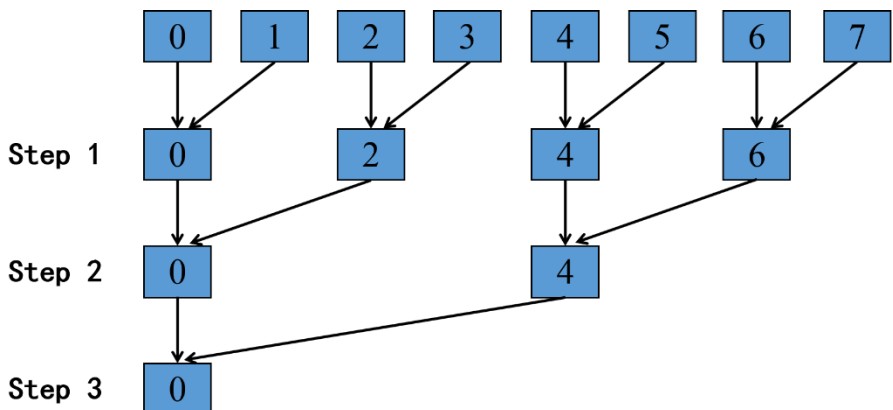

**Figure 7.** Register communication.

Step 1: The data of CPEs 1, 3, 5, and 7 are transmitted to the CPEs 0, 2, 4, and 6 through register communication.
Step 2: The data of CPEs 2 and 6 are transmitted to the CPEs 0 and 4 through register communication.
Step 3: The data of CPE 4 are transmitted to the CPE 0 through register communication.
Step 4–Step 6: Similar to Step 1 to Step 3, Step 4 to Step 6 operate on the column, and finally all the data that needs to pass will be transmitted from 64 CPEs to CPE 0.

Step 7: CPE 0 communicates with the main memory by DMA.

With this approach, CPE 0 is mainly responsible for the communication between the CPEs and MPE, so CPE 0 will be responsible for fewer computation tasks than other CPEs to achieve the purpose of reducing the communication time. Since register communication is much faster than access to the main memory, this method can greatly reduce the communication time.

## 4. Experiment and Results

### 4.1. Experimental Environment

The Sunway TaihuLight computer system uses the SW26010 heterogeneous many-core processor, using the 64-bit independent SW (Sunway) instruction set, full-chip 260 cores, a chip standard operating frequency of 1.5 GHz, and a peak computing speed of 3.168 TFLOPS. The Sunway TaihuLight high-speed computing system has a peak computing speed of 125.436 PFLOPS, a total memory capacity of 1024 TB, a total memory access bandwidth of 4473.16 TB/s, a high-speed interconnection network halved bandwidth of 70 TB/s, and an input/output (I/O) aggregation bandwidth of 341 GB/s. The measured LINPACK continuous operation speed is 93.015 PFLOPS, the LINPACK efficiency is 74.153%, the system power consumption is 15.371 MW, the performance power consumption ratio is 6051.131 MFLOPS/W, the auxiliary computing system's peak operation speed is 1.085 PFLOPS, the total memory capacity is 154.5 TB, and the total disk capacity is 20 PB.

The Sunway TaihuLight computer system supports parallel programming models that are internationally integrated, including MPI3.0, OpenMPI3.1, Pthreads, and OpenACCess2.0, and it also supports message parallel programming models, shared parallel programming models, and accelerated parallel programming models. It also provides an accelerated thread library programming interface customized for the structural characteristics of the SW26010 heterogeneous many-core processor.

### 4.2. Numerical Experimental Results

#### 4.2.1. Parallel Efficiency

In order to test the acceleration effect under different computations, this paper selects the grid size of 50 million, 100 million, and 200 million; the maximum number of MPEs is 2048, and the maximum total number of cores is 133,120. The acceleration effects of only the MPEs and the MPEs plus CPEs mode are counted separately.

Figure 8 shows the speedup and parallel efficiency for only MPEs operations at three problem sizes. Figure 9 shows the percentage of communication time spent on only MPEs for the three problem sizes. As can be seen in Figure 8, both the speedup ratio and the efficiency decrease as the number of computational cores increases. This is because, for the same problem size, as the number of computational cores increases, the number of computational grids on each core decreases. For one computational core, if we assume the size of the computation area to be $NX \times NY$, the number of grid points that the current computation core needs to send data to the adjacent computation core is $NX \times NY - (NX - 2) \times (NY - 2)$, and the ratio of the grid points that need to send data to the total grid point of the calculated core is R, where

$$R = \frac{NX \times NY - (NX - 2) \times (NY - 2)}{NX \times NY} = \frac{2(NX + NY + 2)}{NX \times NY}. \tag{3}$$

It can be seen from Equation (3) that, as the number of computational cores increases, both NX and NY decrease, and the ratio R increases. That is, the percentage of communication time in total time increases (Figure 9). Therefore, parallel efficiency will gradually decline.

Similarly, for the same number of computational cores, as the problem size increases, the amount of computational grid allocated to each computational core increases, and the percentage of communication time as a percentage of total time decreases (Figure 9); therefore, the acceleration ratio

and parallelism efficiency will increase (Figure 8). When the problem size is 200 million, the parallel efficiency is over 94%.

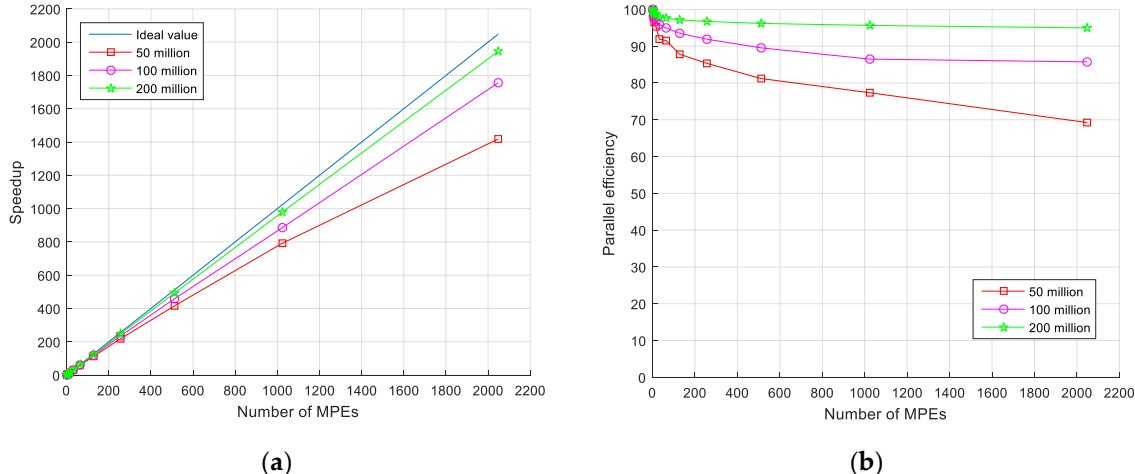

(**a**)                (**b**)

**Figure 8.** For only management processing elements (MPEs) at three problem sizes: (**a**) Acceleration ratio; (**b**) Parallel efficiency.

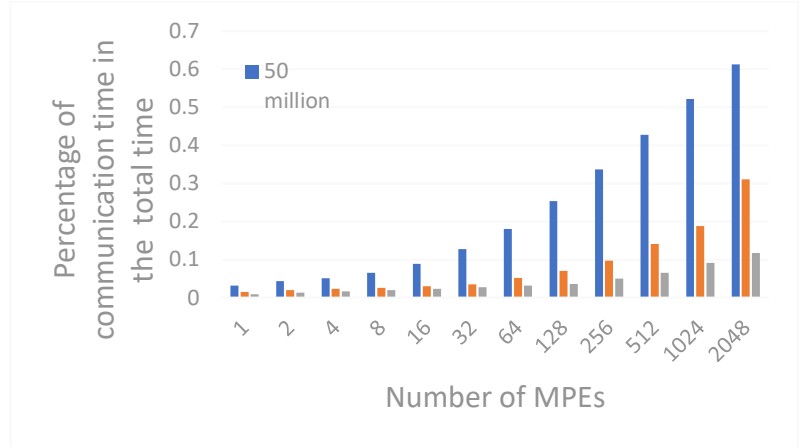

**Figure 9.** Communication time as a percentage of total time for only MPEs at three problem sizes.

As the problem size is 200 million, only MPEs have the highest parallel efficiency. Therefore, this paper uses the 200 million problem size to test the acceleration effect of the CPEs, and tests up to 133,120 cores.

Based on a single CG (1 MPE and 64 CPEs, a total of 65 cores), the number of CGs is increased gradually. The acceleration ratio and parallel efficiency are shown in Table 2 and in Figure 10a. The Mega Lattice Site Updates Per Second (MLUPS) is shown in Figure 10b. As the number of CGs increases, the parallel efficiency and MLUPS decrease, and the speedup and parallel efficiency also decrease compared with the only MPE mode. This is mainly attributed to the startup and shutdown of the CG operation thread group and the communication between the MPE and CPEs. However, when the number of CGs is 2048 (133,120 cores total), the efficiency can still reach more than 72%, and our simulation performs at 285,600 MLUPS on 2048 CGs, corresponding to 54.8% of the effective memory bandwidth performance.

**Table 2.** Acceleration ratio and efficiency based on a single core-group (CG) with the highest number of 2048 CGs (a total of 133,120 cores).

| CGs | 1 | 2 | 4 | 8 | 16 | 32 | 64 | 128 | 256 | 512 | 1024 | 2048 |
|---|---|---|---|---|---|---|---|---|---|---|---|---|
| Cores | 65 | 130 | 260 | 520 | 1040 | 2080 | 4160 | 8320 | 16640 | 33280 | 66560 | 133120 |
| Theoretical acceleration ratio | 1 | 2 | 4 | 8 | 16 | 32 | 64 | 128 | 256 | 512 | 1024 | 2048 |
| acceleration ratio | 1 | 1.9983096 | 3.9549608 | 7.84065056 | 15.6233262 | 30.9049656 | 60.5670289 | 117.7244635 | 228.170942 | 440.4674223 | 841.5104023 | 1492.812774 |
| Parallel efficiency (%) | 100 | 99.915482 | 98.874021 | 98.0081321 | 97.645789 | 96.5780176 | 94.6359827 | 91.9722371 | 89.1292742 | 86.02879342 | 82.17875022 | 72.89124874 |

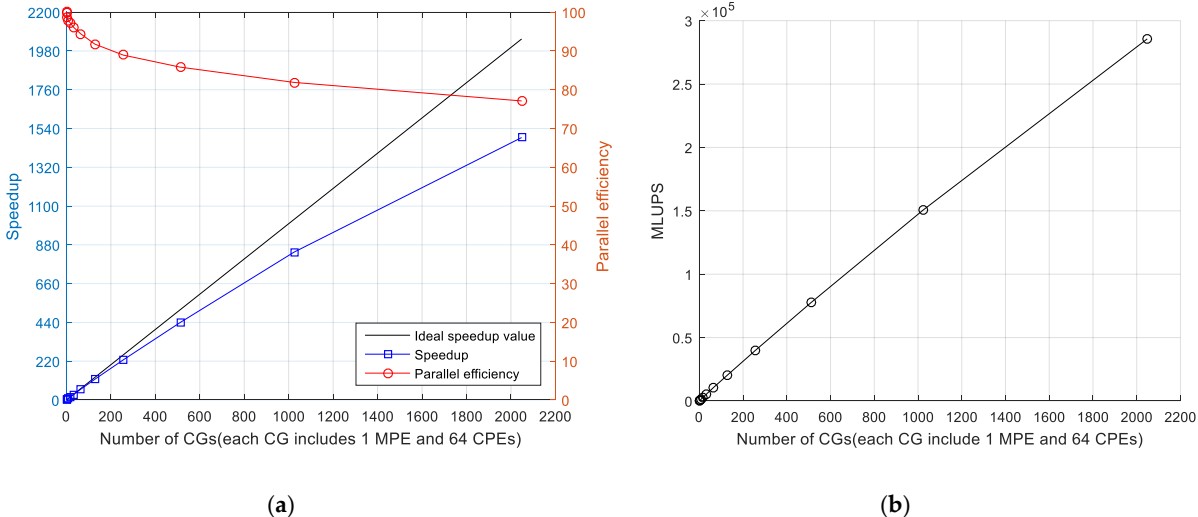

(**a**) (**b**)

**Figure 10.** (**a**) Acceleration ratio and efficiency with computing processing elements (CPEs); (**b**) Mega Lattice Site Updates Per Second (MLUPS) with CPEs.

### 4.2.2. Airfoil Computation Results

The computation area when calculating the flow around the 2D NACA0012 airfoil is shown in Figure 11. *C* is the airfoil length, and C = 400*dx*, where *dx* is the grid size. The entire computation comprises 200 million grids. A velocity boundary condition and non-equilibrium extrapolation boundary condition are respectively used on the inlet boundary and outlet boundary. The standard bounce back boundary condition is used for the upper, lower, and airfoil edges [14]. The inflow Mach number M = 0.1, the Reynolds number Re = 1000, and a total of 1 million time steps are calculated; and the angles of attack are calculated as 0°, 4°, 6°, 8°, 10°, 12°, 15°, 17°, 20°, 22°, 25°, and 28°.

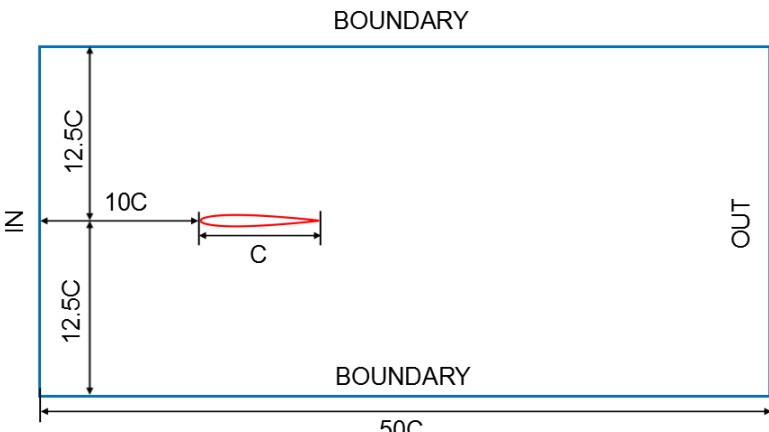

**Figure 11.** Computation area. The left side is the inflow and the right side is the outflow.

Figures 12 and 13 show the pressure and velocity clouds at several representative angles of attack with angles of attack of 0°, 4°, and 8° and 12°, 15°, and 20°, respectively. It can be seen from Figure 12 that the flow tends to be stable for angles of attack of 0° and 4°, and there is no flow separation phenomenon, which is consistent with the literature [21–23]. When the angle of attack is 8° or higher, unsteady vortex shedding occurs after the airfoil tail, and, as the angle of attack increases, the vortex shedding becomes more obvious than before and the flow instability becomes larger. It can also be seen from Figures 12 and 13 that, with an increase in the angle of attack, the pressure difference between the lower side and the upper side of the airfoil increases gradually; therefore, it can be qualitatively known that the lift coefficient will also increase, which can be understood as the vortex lift generated by the shedding vortex.

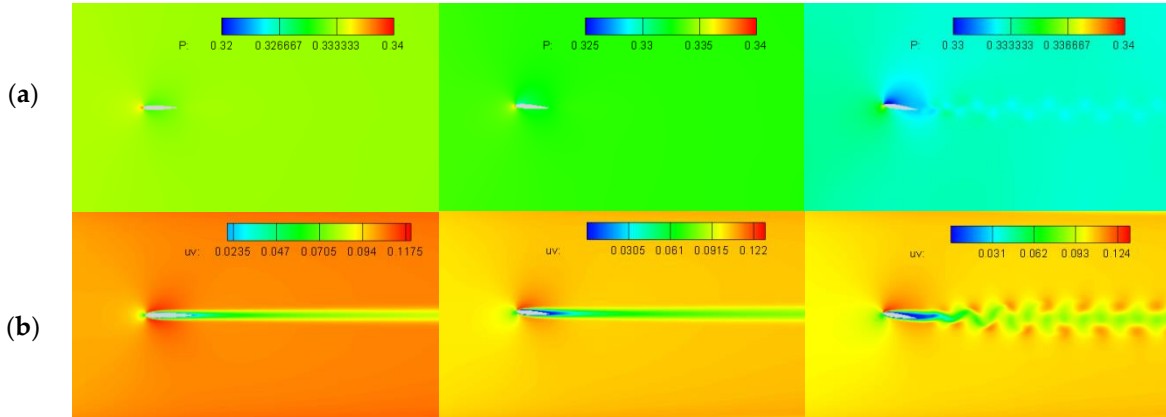

**Figure 12.** Pressure and velocity clouds at different angles of attack: (**a**) pressure distribution on the upper side; (**b**) lower right side of the foil. The angles of attack are 0°, 4°, and 8° (left to right).

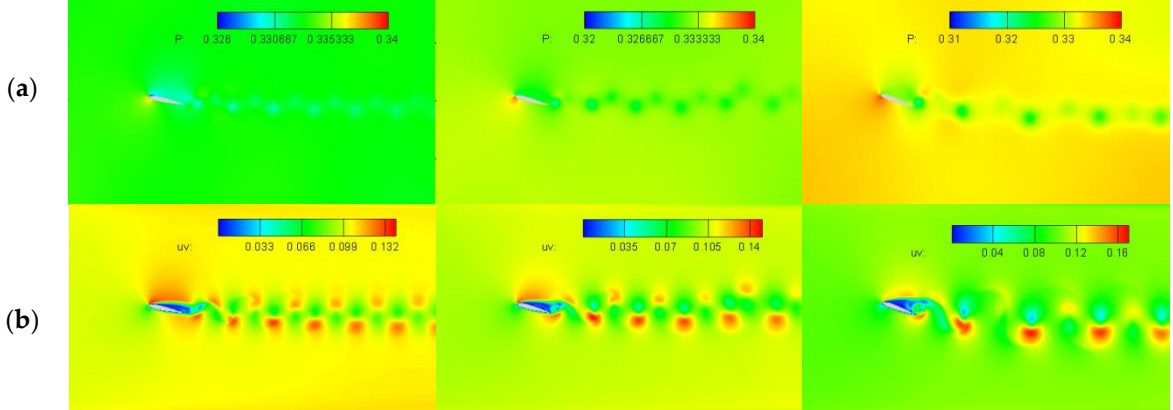

**Figure 13.** Pressure and velocity clouds at different angles of attack: (**a**) pressure distribution on the upper side; (**b**) velocity distribution on the lower right side of the foil. The angles of attack are 12°, 15°, and 20° (left to right).

In order to further quantify the main characteristics of the flow and evaluate the aerodynamic performance of the airfoil, the lift coefficient $C_L$ and the pressure coefficient $C_p$ of the upper and lower surfaces of the airfoil are calculated, respectively, as

$$C_L = \frac{F_L}{\frac{1}{2}\rho_0 u_0^2 C}, \qquad C_p = \frac{p - p_0}{\frac{1}{2}\rho_0 u_0^2},$$

where the subscript 0 represents the value under far-field conditions, respectively, $C$ is the size of the airfoil, and $F_L$ is the lift, i.e., the vertical component of the aerodynamic force acting on the airfoil chord.

Figure 14a shows the average pressure coefficient distribution of the upper and lower surfaces when the angle of attack is 8°. Owing to the angle of attack, the average pressure coefficients for the upper and lower surfaces are different at the front of the airfoil, and they gradually change with the position; this appears to be a consistent trend. The trend and the extreme value of the computation result are in good agreement with those in the literature [22], and a mean error less than 1.5%. Figure 14b shows the trend of the lift coefficient as a function of the angle of attack. This result is also in good agreement with the results in the literature [24], and a mean error less than 5%. Once the angle of attack is greater than 25°, the vortex may be broken, and the lift coefficient is reduced; this result agrees well with the critical angle of the stall phenomenon in [25].

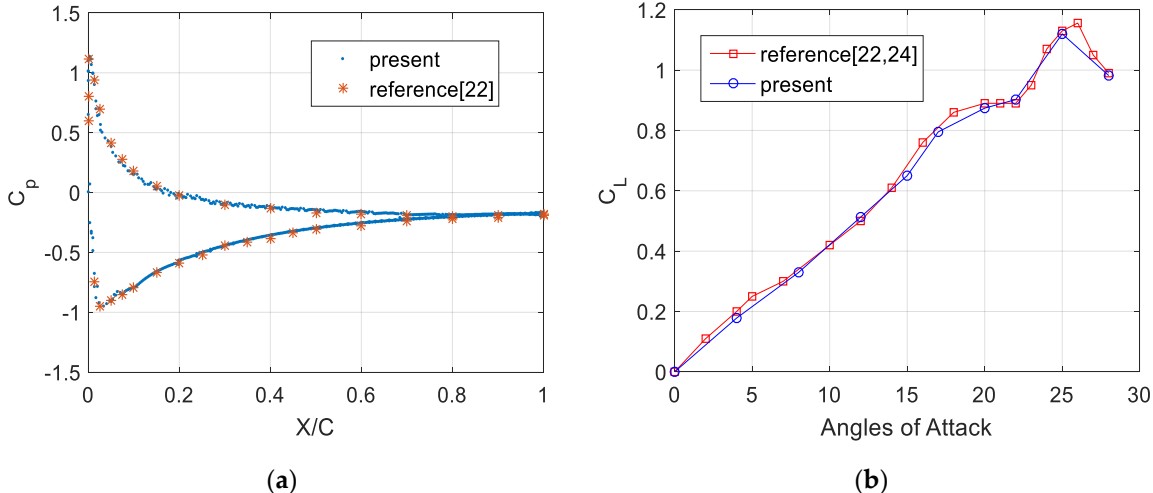

**Figure 14.** (**a**) Average pressure coefficient ($C_p$) distribution at an angle of attack of 8°; (**b**) The trend of the lift coefficient ($C_L$) with the angle of attack.

## 5. Conclusions

This paper discussed the implementation and optimization of the LBM based on the Sunway TaihuLight SW26010 heterogeneous multi-core processor architecture and the simulation computation of NACA0012 2D airfoil flow. In parallel computing, the focus is on two-layer parallel design, the parallelism between CGs, and the parallelism between MPEs and CPEs. Parallelism between CGs is achieved by MPI, and that between MPEs and CPEs is achieved by the accelerating thread library. We adopted the double-buffer mode to reduce the communication time and adjust the discrete storage structure along with other optimization methods; these achieved a good parallel efficiency and speedup ratio for the Sunway TaihuLight. In terms of airfoil computation, the pressure and velocity distributions at different angles of attack were calculated respectively; and the average pressure coefficient and lift coefficient were obtained for an angle of attack of 8°. The results were in good agreement with the literature.

Future work will focus on optimizing procedures, improving efficiency—especially the efficiency of CPEs—exploring 3D models, more large-scale numerical computation of airfoils, and increasing the number of MPEs to one million. We will also explore the application of the LBM in flow noise. A common method for solving flow noise is to solve the Ffowcs Williams and Hawkings (FW-H) equation. High accuracy of the flow field information is required for solving the FW-H equation. The LBM is a discrete format of the Boltzmann equation. Computation results from the LBM can reflect the fine structure of the flow field and provide accurate numerical results for the computation of aerodynamic noise. Long-term research shows that the LBM is indeed suitable for the computation of aerodynamic noise [26,27].

**Author Contributions:** Conceptualization, L.W. and W.Z. (Wu Zhang); Formal analysis, X.Z.; Funding acquisition, W.Z. (Wenhao Zhu) and W.Z. (Wu Zhang); Investigation, L.W.; Methodology, L.W., W.W. and X.C.; Project administration, W.Z. (Wu Zhang); Resources, W.Z. (Wenhao Zhu) and K.X.; Supervision, L.W.; Validation, K.X.; Visualization, W.W.; Writing—original draft, L.W.; Writing—review & editing, L.W.

**Funding:** This research was funded by the National Natural Science Foundation of China major research project key project (91630206).

**Acknowledgments:** Firstly, we would like to thank Wu Zhang, Weiguo Wu and Wenhao Zhu for providing us with the basic theoretical knowledge of LBM and financial support. We learned a lot from Kangle Xu about airfoils flow and aerodynamics. We express our sincere thanks to Xuesen Chu for his help in achieving better parallel performance on the Sunway TaihuLight supercomputer. Finally,we should express our gratitude to Liangjun Wang and Xiaoxiao Zhang for the implementer of the experiment and writing.

**Conflicts of Interest:** The authors declare no conflict of interest.

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
