# Peer review of "Accurate Computation of Airfoil Flow Based on the Lattice Boltzmann Method"

_applsci, doi:10.3390/app9102000_

Round 1
Author Response
dear reviewer:
Thank you for your confident review. Your comments are very valuable. I have answered and explained your questions one by one(see the file "response to reviewer 1.pdf"). Thanks again.
authors

Reviewer 2 Report
The authors present performance and scalability results for a 2D Lattice Boltzmann simulation on Sunway Taihunlight.
While the method LB in 2D is well-known to scale and the presented model itself is far from new, the presented work is definitely valuable for Sunway users and programmers of this kind of hardware.
The performance numbers are convincing in this revised version of the manuscript.
I can recommend publication of the paper, after the following minor comments have been addressed.
Comments:
- Abstract: "The proposed model provides accurate flow field data for performing further computations of aerodynamic noise."
-> There is not enough proof that the order of accuracy for aeroacoustics is sufficient and it is not related to the presented work so I would suggest to remove this sentence
- L. 26-28: three subsequent sentences talk about "important technical mechanism", "important research field", "important numerical simulation method". Maybe rephrase to get rid of "important" once or twice?
- L. 50: Remove "A." in "A.Lintermann"
- I do not understand the citation of [4]. The paper does not cope with new LB formulations and alternative forms of distribution functions. Moreover, the work by the Rude group is actually more famous for scalability and performance which is the essential brick in this work. So I would suggest to change the citation here and better refer to one of the group's scalability works.
- I would remove, similar to my comment on the abstract, the paragraph L.55-60, and move this discussion to the conclusions and outlooks section.
- L. 72: Sentence "Supercomputers have become.." is superfluous and can be removed.
- L. 89: change "the Boltzmann equation" to "the Lattice Boltzmann equation" as the following equation--as the authors point out--is already the discrete version.
- L. 120: change "divide the task of the mesh" to "partition the mesh"
- L. 122: change to "2D partitioning is ...than 1D partitioning"
- Figure 3: change "partition" to "partitioning" for all occurences in caption
- Figure 5 is rather a listing or simply code snippet than a figure.
- L. 161: "adjust the neighbor... and adjust the discrete" -> this sentence does not make sense, probably the authors mean "the neighbor... and the discrete storage... need to be adjusted"?
- Concerning the memory adjustment in L. 161 and following: is this step performed separately for each grid point, or did the authors consider to apply blocking (i.e. do the operation for several grid points)? The latter could be faster, I guess?
- L. 209ff: Figure 6 and 7 should be probably Figure 9 and 10?
- "Scale of computation" or "computation scale" is not such a well-known term. Better use "problem size".
- For sake of transparency of the results, I suggest to include one sentence after/before line 245, stating "our simulation performs at X MLUPS on a single CG, corresponding to Y% of the effective memory bandwidth performance" and mention again that LB should be mem bound.
- L. 253: where C = 400 grid -> please correct the structure of this sentence
- L. 253: change to "The entire computation comprises 200 million... . A velocity boundary condition... and non-equilibrium...conditions are used..."
- L. 293ff: it is mentioned "good agreement" of values for several quantities that have been studied. The authors should quantify this (how much % difference of the values).
- Figure 15: The authors should double-check that the references 21,22 refer to the correct sources in this figure. I guess they wanted to refer to 24,26 or 27?
Author Response
dear reviewer:
Thank you for your confident review. Your comments are very valuable. I have answered and explained your questions one by one(see the file "response to reviewer 2.pdf"). Thanks again.
authors

This manuscript is a resubmission of an earlier submission. The following is a list of the peer review reports and author responses from that submission.
Round 1
Reviewer 1 Report
The manuscript discusses a 2D Lattice Boltzmann implementation which is executed on the Sunway Taihulight supercomputer. It further discusses some implementational aspects of the work, provides scalability graphs which show good scalability and further provides some validation for flow around an airfoil.
The language is mostly OK, with some typos and inconsistencies listed below (minor points). The methodology used for LB is not new but around for 20 years or so.
The actual interesting part would be the specific implementation on the pretty novel Sunway architecture; however, it is exactly this part which is kept very generic without any information for LB users and developers. Besides, although scalability looks good, there is no proof that this is really efficient code (see my comment on sequential performance; without this measure, the scalability discussion is rather weak).
I can therefore not recommend a publication in the current form; more details reasons for this are:
- There are no references on HPC implementations and corresponding efforts for LB codes. The authors should include an additional paragraph on this topic in the introduction.
- Section 3.3 could be very interesting. However, there is basically no mentioning, how the describe methodology actually translates to the used LB algorithm. For example, in Figure 4, how is this mapped to LB quantities and the actual LB implementation? Do you do standard AB-pattern or AA-pattern? How do you store and allocate your data (SoA vs AoS) etc?
- Sec. 4.1: there is much said about the Sunway Taihulight and related programming paradigms, but it is not mentioned which of these paradigms and libraries have been used in the present work. A section on "Implementation" for the present work would be highly desirable.
- Sec. 4.2.1: the simulations are carried out on up to 133k cores, which is a fraction of 1% of the full Sunway machine. While I really appreciate the efforts for doing this and there is no doubt that this massive parallelism is difficult to cope with, there are, yet, several works that have already accomplished similar simulations, even in 3D with complex physics. See amongst others the works by the waLBerla research group, or https://www.sciencedirect.com/science/article/pii/S0032591018309501 for work on the Sunway machine.
- Sec. 4.2.1: the authors argue that scalability decreases due to increasing communication cost. This is no arguing, since it is basically ALWAYS that scalability decreases due to this. It would be more relevant to actually provide performance numbers and metrics that show (a) how good the sequential LB implementation behaves, and then (b) relate this to the scalability curve. For example, if the sequential LB implementation is incredibly slow, scalability will always be good, especially for strong scaling as carried out in this work. I say this because I find it surprising that a 2D LB implementation scales up to basically 2000 grid cells per compute core (which is not that much) in a 133k core setup. I suggest to provide sequential performance numbers and scalability plots, for example using the LB metrics "Lattice updates per second" and relate this further to the Sunway hardware characteristics (particularly relevant bandwidth values) to actually show how close the code runs with regard to peak performance.
- Sec. 4.2.2: What kind of boundary condition is used at the outlet of the simulation (i.e. on the right side)?
Minor points:
- Introduction: The authors argue that high accuracy is relevant for acoustics simulations, but the paper does not deal with any acoustics simulation at the end. I would therefore suggest to only introduce acoustics as side motivation.
- L. 26: "super computer" -> supercomputer
- L. 36: "make enable" -> enable
- L: 53: "based on the platform's,..." -> some word is missing here behind "platform's"
- L. 59f: "In terms of aeronautical-engineering computations, the wing is the main device for generating lift and its structure is fine, which has become the focus of research"
-> I do not understand this sentence, in particular the part " its structure is fine"??? Please re-phrase!
- L. 95: "After partition, the cartesian" -> After partitioning the Cartesian
- L. 99: " it is a fluid poin" -> it is a fluid point
- L. 104: "1D partition and 2D partition" -> 1D partitioning and 2D partitioning
- L. 212: "standard bounce boundaries" -> standard bounce back boundaries; please also include relevant references for the boundary treatment in this line and the ones before
- Figure 11: please include the corresponding reference in the caption which is used as "reference" value in the plots
Author Response
dear reviewer:
Thank you very much for your careful review. Your comments are very useful to me. I have corrected and replied to the relevant questions one by one.
Liangjun Wang

Reviewer 2 Report
This paper presents a 2D study of the flow over an airfoil. Before recommending the paper for publication, I have some concerns.
The description of the LBM is sloopy. In line 81, which is the value of c_s? 1/sqrt(3) I suppose. In Eq.1, which is the form of F_alpha? How are density and velocity computed? Do the authors use interpolated bounce-back? Half-way bounce-back? Full-way bounce-back?
Finally, the LBM literature shows a rich variety of studies about flapping wing dynamics. I suggest to have a look at:
- Computers & Structures 153, 230-238, 2015
- Journal of Fluids and Structures 49, 516-533, 2014
- International Journal for Numerical Methods in Fluids 83.4 (2017): 331-350
- Fluid Dynamics Research 49.5 (2017): 055504
among the others.
Author Response

(The authors gave the same response as above.)

Round 2
Reviewer 1 Report
Some of my comments have been addressed, but not all of them. I will go through the remaining points step by step.
=============
Point 3: There are no references on HPC implementations and corresponding efforts for LB
codes. The authors should include an additional paragraph on this topic in the introduction.
Response 3: I have added the relevant introduction in the second paragraph of the
introduction
ANSWER: There have been some references included; yet, these are not representative for HPC and LB. There have been works, for example by the groups of Ulrich Rüde / Manfred Krafczyk / Wellein, Hager that have reported much progress on HPC implementations on LB. Other works related to the topic comprise GPU implementations by Obrecht et al., just to name a few references that should be checked and potentially cited in a reasonable way!
And I had also listed LB works that had been carried out on the Sunway machine already in the first review round!!!
============
Point 4: Section 3.3 could be very interesting. However, there is basically no mentioning,
how the describe methodology actually translates to the used LB algorithm. For example, in
Figure 4, how is this mapped to LB quantities and the actual LB implementation? Do you do
standard AB-pattern or AA-pattern? How do you store and allocate your data (SoA vs AoS)
etc?
Response 4: Figure 5 shows a detailed description of how the double-buffering mode is
applied to LBM code. We used the SoA.
ANSWER: The figure 5 does not show any kind of typical LB steps, i.e. collision/streaming. While this is OK as there is always a gap between algorithmics and actual HPC implementation, I would expect at least some useful descriptions that make readers bridge this gap and make them understand where/how LB comes into play in this buffering scheme.
==============
Point 6: The simulations are carried out on up to 133k cores, which is a fraction of 1% of the
full Sunway machine. While I really appreciate the efforts for doing this and there is no doubt
that this massive parallelism is difficult to cope with, there are, yet, several works that have
already accomplished similar simulations, even in 3D with complex physics. See amongst
others the works by the waLBerla research group, or
https://www.sciencedirect.com/science/article/pii/S0032591018309501 for work on the
Sunway machine.
Response 6: Your opinion is very correct. LBM has been developed for so many years. It has
indeed had a lot of excellent research results. Our work really only uses a small part of the
core of Shenwei, but our work is a continuous study. On this basis, we will continue to study
and increase the number of cores to make the calculation more accurate and provide a data
foundation for the subsequent calculation of aerodynamic noise. The current work will be a
solid foundation for our follow-up work. A lot of work, although the calculation methods are
similar, but the platforms used are different, so the methods implemented are also different.
To be a long-term research, a fixed platform is very important, so we feel that our work is
still valuable.
ANSWER: I still would expect to get this and further motivation when I read the paper. Can you include this in the introduction or conclusion and point out those aspects that will make your work differ from others?
======================
Response 7: Thank you for your suggestion. In a 133k core setup, there is only 2000 grid
cells per compute core and that is indeed not nuch. I did this to make the calculation faster for
single CG, because the total grid cell is the same regardless of the number of CGs. In 4.2.1 I
added a figure(Figure8(b)) to show The Mega Lattice Site Updates Per Second (MLUPS) .
ANSWER: Looking at the MLUPS, this does not appear to be much. The performance numbers from the new plot suggest O(50) MLUPS per CG, i.e. using 1 MPE and 64 CPEs. Each CG has to my knowledge O(140) GB/s bandwidth. Pumping through two D2DQ9 arrays (each cell with 2x9 doubles) and considering O(50) MLUPS, this corresponds to 50*2*9*8 million bytes/second=7GB/s which is a 5% fraction of the actual bandwidth; and thus, being typically bandwidth bound, the LB runs at 5% of the actually possible performance, correct?
The authors should please double-check this computation and give a reasonable explanation/interpretation of it, and correspondingly address this issue in the manuscript.
Author Response
dear reviewer:
Thank you for your confident review. Your comments are very valuable. I have answered and explained your questions one by one.Thanks again.
authors

Round 3
Reviewer 1 Report
All points have been addressed sufficiently, except for a direct mapping of the buffering on the streaming step notation, and the following, most important point:
As expected, the performance provided here is really low, that is 5% of what should actually be achievable. The authors agrees on this, as can be seen in the last report; however, it is not only about reaching the 5% at 130,000 cores, but even at single-node level at moderate amount of cores.
I therefore vote to reject the report, but would strongly suggest and appreciate a resubmission of this work, when the LB simulation has been tuned to leverage more performance and then resubmit it to ApplSci. This will then be definitely a good and very valuable work!